# Effects of teriparatide and bisphosphonate on spinal fusion procedure: A systematic review and network meta-analysis

**Shih-Hao Cheng[1,2], Yi-Jie Kuo[1,3], Chiehfeng Chen[4,5,6,7], Yi-No Kang[5,7,8,9]***

**1** Department of Orthopedics, Wan Fang Hospital, Taipei Medical University, Taipei, Republic of China (Taiwan), **2** Department of Orthopedics, Cheng Hsin General Hospital, Taipei, Republic of China (Taiwan), **3** Department of Orthopedic Surgery, School of Medicine, College of Medicine, Taipei Medical University, Taipei, Republic of China (Taiwan), **4** Department of Public Health, School of Medicine, College of Medicine, Taipei Medical University, Taipei, Republic of China (Taiwan), **5** Cochrane Taiwan, Taipei Medical University, Taipei, Republic of China (Taiwan), **6** Division of Plastic Surgery, Department of Surgery, Wan Fang Hospital, Taipei Medical University, Taipei, Republic of China (Taiwan), **7** Evidence-Based Medicine Center, Wan Fang Hospital, Taipei Medical University, Taipei, Republic of China (Taiwan), **8** Research Center of Big Data and Meta-Analysis, Wan Fang Hospital, Taipei Medical University, Taipei, Republic of China (Taiwan), **9** Institute of Health Policy & Management, College of Public Health, National Taiwan University, Taipei, Republic of China (Taiwan)

* academicnono@gmail.com

**Data Availability Statement:** The data underlying the results presented in the study are available from relevant studies: 1. Chen F, Dai Z, Kang Y, Lv G, Keller ET, Jiang Y. Effects of zoledronic acid on

## Abstract

### Background

Giving patients anti-osteoporotic agents peri-operatively is a well-accepted strategy to increase fusion rate and prevent complications. The purpose of this study was to investigate effectiveness of teriparatide and bisphosphonate on fusion surgery of thoracic and lumbar spine.

### Methods

We searched EMBASE and PubMed for randomized clinical trials (RCTs) and prospective comparative studies using teriparatide or bisphosphonate in peri-operative spinal fusion surgery. Our synthesized data of fusion rate, Oswestry disability index (ODI), and adverse event in contrast-based network meta-analysis. Pooled results were presented in risk ratio (RR) or mean difference (MD) with 95% confidence interval (CI).

### Results

Our search hit eight RCTs and three prospective studies with 676 patients receiving spinal surgery. Pooled result showed that teriparatide+Denosumab leads to significantly higher fusion rate than placebo (RR, 2.84; 95% CI: 1.22 to 6.60) and bisphosphonate (RR, 2.59; 95% CI: 1.13 to 5.96). We did not observe significant finding among placebo, teriparatide, and bisphosphonate in the two network models.

bone fusion in osteoporotic patients after lumbar fusion. Osteoporosis International. 2016;27 (4):1469-76. doi: 10.1007/s00198-015-3398-1. 2. Cho PG, Ji GY, Shin DA, Ha Y, Yoon DH, Kim KN. An effect comparison of teriparatide and bisphosphonate on posterior lumbar interbody fusion in patients with osteoporosis: a prospective cohort study and preliminary data. European Spine Journal. 2017;26(3):691-7. doi: 10.1007/s00586-015-4342-y. PubMed Central PMCID: PMCLilly (United States). 3. Ebata S, Takahashi J, Hasegawa T, Mukaiyama K, Isogai Y, Ohba T, et al. Role of weekly teriparatide administration in osseous union enhancement within six months after posterior or transforaminal lumbar interbody fusion for osteoporosis-associated lumbar degenerative disorders: A multicenter, prospective randomized study. Journal of Bone and Joint Surgery - American Volume. 2017;99(5):365-72. doi: 10.2106/JBJS.16.00230. 4. Ide M, Yamada K, Kaneko K, Sekiya T, Kanai K, Higashi T, et al. Combined teriparatide and denosumab therapy accelerates spinal fusion following posterior lumbar interbody fusion. Orthopaedics and Traumatology: Surgery and Research. 2018;104(7):1043-8. doi: 10.1016/j.otsr.2018.07.015. 5. Jespersen AB, Andresen ADK, Jacobsen MK, Andersen MO, Carreon LY. Does Systemic Administration of Parathyroid Hormone After Noninstrumented Spinal Fusion Surgery Improve Fusion Rates and Fusion Mass in Elderly Patients Compared to Placebo in Patients With Degenerative Lumbar Spondylolisthesis? Spine (Phila Pa 1976). 2019;44(3):157-62. Epub 2018/07/14. doi: 10.1097/brs.0000000000002791. PubMed PMID: 30005049. 6. Li C, Wang HR, Li XL, Zhou XG, Dong J. The relation between zoledronic acid infusion and interbody fusion in patients undergoing transforaminal lumbar interbody fusion surgery. Acta Neurochirurgica. 2012;154(4):731-8. doi: 10.1007/s00701-012-1283-7. 7. Nagahama K, Kanayama M, Togawa D, Hashimoto T, Minami A. Does alendronate disturb the healing process of posterior lumbar interbody fusion? A prospective randomized trial. Journal of Neurosurgery: Spine. 2011;14(4):500-7. doi: 10.3171/2010.11.SPINE10245. 8. Ohtori S, Inoue G, Orita S, Yamauchi K, Eguchi Y, Ochiai N, et al. Comparison of teriparatide and bisphosphonate treatment to reduce pedicle screw loosening after lumbar spinal fusion surgery in postmenopausal women with osteoporosis from a bone quality perspective. Spine. 2013;38(8):E487-E92. doi: 10.1097/BRS.0b013e31828826dd. PubMed Central PMCID: PMCEisai(Japan) Lilly(Japan). 9. Seki S, Hirano N, Kawaguchi Y, Nakano M, Yasuda T, Suzuki K, et al. Teriparatide versus low-dose bisphosphonates before and after surgery for adult

## Conclusion

This is the first network meta-analysis providing an overview of the use of teriparatide and bisphosphonate for spinal fusion surgery. Teriparatide treatments are worth to be consider for spinal fusion surgery.

## Introduction

Spinal fusion surgery is widely employed to treat lumbar stenosis, instability, intervertebral disc degeneration, deformity, and trauma of spine. In recent years, the number of patients undergoing the operation has been increasing in USA [1, 2]. The stability of spinal instrumentation relies on good bone quality and the pullout strength of pedicle screws is highly correlated with bone mineral density of spine [3]. Among patients undergoing the procedure, many are geriatrics with high prevalence of low bone mass and osteoporosis [1, 4]. Operation on an osteoporotic spine increases risk of complications, such as implant migration, instrumentation failure, adjacent compression fracture and pseudoarthrosis [5]. According to previous research, pseudoarthrosis after lumbar spine fusion is not rare, ranging from 5% to 35% [6, 7], and may be associated with poor outcomes [8]. As a result, successful treatment of osteoporosis and elevated fusion after spine surgery are important and challenging issues for surgeons. Giving patients anti-osteoporotic agents peri-operatively is a well-accepted strategy to increase fusion rate and prevent complications [9]. Among many anti-osteoporotic medicines, teriparatide [recombinant human PTH (1–34)] and bisphosphonate are most commonly used [9]. Teriparatide is an artificial synthetic parathyroid hormone. When given intermittently, it exhibits strong anabolic effect on skeleton and increases bone mineral density by stimulating new bone formation [10]. Bisphosphonate, on the other hand, is a derivative of inorganic pyrophosphate. It inhibits the activity of osteoclast and increases bone stock by decreasing bone absorption [11]. While there are studies supporting the use of both agents [12, 13], pooled evidence on their comparative efficacy is still lacking. To make up for such deficiency, this study performed a network meta-analysis to synthesize the current evidence on effectiveness of teriparatide and bisphosphonate in fusion surgery of thoracic and lumbar spine.

## Methods

According to the Cochrane handbook and PRISMA guidelines, eligibility criteria were defined for this comprehensive synthesis and search strategy was developed. Databases and reference lists of relevant studies were searched and evidence was selected, followed by data extraction and quality assessment. The consistency model was then formed and network meta-analysis was conducted. This network meta-analysis was exempted from institutional review board approval because this study synthesized and analyzed only published data.

### Evidence selection criteria

According to the proposed research question, this comprehensive synthesis selected evidence if (a) the study recruited patients undergoing spinal fusion; (b) the intervention involved the administration of teriparatide or bisphosphonate; and (c) the study design was randomized clinical trial or prospective comparative investigation with two or more arms. Exclusion criteria defined to enhance the validity of the comprehensive synthesis were as follows: (a) animal study; (b) studies recruited cervical spinal fusion; (c) gray literature without detailed

spinal deformity in female Japanese patients with osteoporosis. European Spine Journal. 2017;26 (8):2121-7. doi: 10.1007/s00586-017-4959-0. 10. Sheng J, Liu D, Zheng W, Zhou JJ, Wu HH, Xu W, et al. Zoledronic acid combined with bone cement augmented pedicle screw for osteoporotic lumbar fusion. Medical Journal of Chinese People's Liberation Army. 2018;43(12):1044-8. doi: 10.11855/j.issn.0577-7402.2018.12.09. 11. Ushirozako H, Hasegawa T, Ebata S, Oba H, Ohba T, Mukaiyama K, et al. Weekly Teriparatide Administration and Preoperative Anterior Slippage of the Cranial Vertebra Next to Fusion Segment < 2 mm Promote Osseous Union After Posterior Lumbar Interbody Fusion. Spine. 2019;44(5):E288-E97. doi: 10.1097/BRS.0000000000002833. 12. Yagi M, Ohne H, Konomi T, Fujiyoshi K, Kaneko S, Komiyama T, et al. Teriparatide improves volumetric bone mineral density and fine bone structure in the UIV+1 vertebra, and reduces bone failure type PJK after surgery for adult spinal deformity. Osteoporosis international: a journal established as result of cooperation between the European Foundation for Osteoporosis and the National Osteoporosis Foundation of the USA. 2016;27(12):3495-502. Epub 2016/06/28. doi: 10.1007/s00198-016-3676-6. PubMed PMID: 27341809.

**Funding:** The authors received no specific funding for this work.

**Competing interests:** The authors have declared that no competing interests exist.

**Abbreviations:** CI, confidence interval; BMD, bone mineral density; MD, mean difference; ODI, Oswestry disability index; PRISMA, Preferred Reporting Items for Systematic Reviews and Meta-Analyses; RCT, randomized clinical trial; RR, risk ratio; SD, standard deviation; SE, standard error; SUCRA, surface under the cumulative ranking.

information, data, or full text; and (d) studies without results on fusion rate, Oswestry disability index (ODI), bone mineral density (BMD), or overall adverse event.

## Search strategy and study selection

The top two online databases in biomedical science, EMBASE and PubMed (including MEDLINE), were the main sources of this comprehensive synthesis. PubMed provides a well-established platform for building the primary search strategy. Relevant terms of spinal fusion, spinal arthrodesis, teriparatide, and bisphosphonate were identified in both free-text and medical subject heading (Emtree in EMBASE and MeSH in PubMed) with appropriate Boolean operator to form the search strategy. Boolean operator "OR" was used for combining the relevant terms of spinal fusion and spinal arthrodesis. As mentioned above, this analysis aims to provide an overview of the effects of two common anti-osteoporotic medicines (teriparatide and bisphosphonate) on patients undergoing spinal fusion procedure rather than making head-to-head comparison of the two anti-osteoporotic medicines. Thus, "OR" was also employed to combine relevant terms of teriparatide and bisphosphonate, thus increasing search sensitivity. Then, the spinal fusion part was combined with the anti-osteoporotic medicine part through Boolean operator "AND". The search strategy developed had no restriction on language and publication date from each database inception until February 07, 2020 (S1 File in S1 Appendix).

After potential references were identified from EMBASE and PubMed, two research members started evidence selection in two steps, namely title and abstract screening and full-text review according to the eligibility criteria previously defined. In case of different judgements on a reference in the individual screening process of evidence selection, a meeting would be called to review the references till a consensus was reached.

## Data extraction and quality assessment

Two research members independently extracted relevant information and outcome data. The relevant information covered the details of study design, location, number of patients, disease, types of spinal surgery, fusion segment, mean age, sex, and treatments. The outcome data included fusion rate, ODI, BMD, and overall adverse event. Fusion rate and overall adverse event rate were binary data, while the other two results were usually continuous data. Events were extracted for binary outcomes, and mean with standard deviation (SD) for continuous outcomes. When the original reports performed only standard error (SE), SD was estimated using the statistical formula $SE = SD/\sqrt{N}$.

According to the relevant information on details of trial design, two research members assessed the quality of each eligible study. About quality assessment of randomized clinical trials, selection bias, performance bias, we assessed detection bias, and attrition bias according to the Cochrane Risk of Bias Tool [14]. Relevant information for quality assessment involved evaluating randomization generation, allocation concealment, blinding (including healthcare providers, participants, and assessors), follow-up duration, loss to follow-up, and analysis type. On the other hand, we used the Newcastle-Ottawa Scale for non-randomized prospective comparative studies [15]. In case of disagreement on quality assessment, another research member would perform the third review and made the final judgement.

## Evidence synthesis and statistical analysis

This study synthesized evidence in both qualitative and quantitative approach. The quantitative approach applied contrast-based network meta-analysis. Fusion rate and adverse event are binary data and hence synthesized in risk ratio (RR). On the other hand, ODI are continuous

data and therefore pooled in weighted mean difference (WMD). Network meta-analysis was conducted using the random-effects model in view of conceptual heterogeneity among eligible evidence. We reported not only effect size (RR or WMD) but also 95% confidence interval (CI). To clarify the effects of teriparatide, bisphosphonate, combination of teriparatide and denosumab, and placebo, the network meta-analysis also performed surface under the cumulative ranking (SUCRA). This statistical method estimates the probability of each group among the most effective groups and shows group ranking of probability. Inconsistency and small study effect in consistency models of fusion rate, ODI, and adverse event rate were also examined, with inconsistency in each result detected using Lu-Ades' loop inconsistency test and small study effects investigated using adjusted funnel plot and Egger's regression intercept. Statistical significance was judged according to the common threshold ($p < 0.05$). All quantitative syntheses we mentioned above were completed using STATA version 14 for Microsoft Windows.

## Results

The developed search strategy obtained 382 references from the EMBASE (n = 231) and PubMed (n = 151), and one reference through reviewing relevant reference lists. After removing 87 duplications from the 383 references, titles and abstracts of the remaining 296 references were reviewed for eligibility. Then, 12 references from three prospective comparative studies and eight randomized clinical trials were eligible for the comprehensive synthesis (Fig 1) [16–27].

### Characteristics of eligibility trials

The abovementioned 11 studies involved 676 patients receiving spinal surgery in China [16, 21, 25], Denmark [20], Japan [18, 19, 22–24, 26, 27], and Korea [17]. These patients were treated with placebo, teriparatide, combination of teriparatide and denosumab, and bisphosphonate. Mean age of patients in each study ranged from 60.7 to 78 years, and there were only 82 males (12.13%) in all these studies. Other details on study design, study location, surgical indication, and follow-up duration are shown in Table 1. S2 File in S1 Appendix displays quality of the studies.

### Fusion rate

A total of nine studies using placebo, teriparatide, combination of teriparatide and denosumab, and bisphosphonate were included in the network meta-analysis of fusion rate (Fig 2A and S3 File in S1 Appendix) [16–22, 24, 25]. Results showed that compared with placebo, teriparatide (RR, 1.26, 95% CI: 1.08 to 1.47) and combination of teriparatide and denosumab (RR, 2.84, 95% CI: 1.22 to 6.60) achieved significantly higher fusion rate, but not bisphosphonate (RR, 1.10, 95% CI: 0.95 to 1.27). SUCRA demonstrated similar trends favoring combination of teriparatide and denosumab (Mean rank = 1; SUCRA = 98.4; Fig 3A and S4 File in S1 Appendix). In view of the closed loop formed by placebo, teriparatide, and bisphosphonate in the network meta-analysis of fusion rate, the loop inconsistency test was performed. The test did not detect inconsistency in the pooled fusion rate within the network model (Chi-square = 0.62, $p = 0.43$; S5 File in S1 Appendix). Moreover, Egger's test did not detect serious small study effects ($t = -0.66$, 95% CI: -1.433 to 0.809, $p = .531$; S6 File in S1 Appendix).

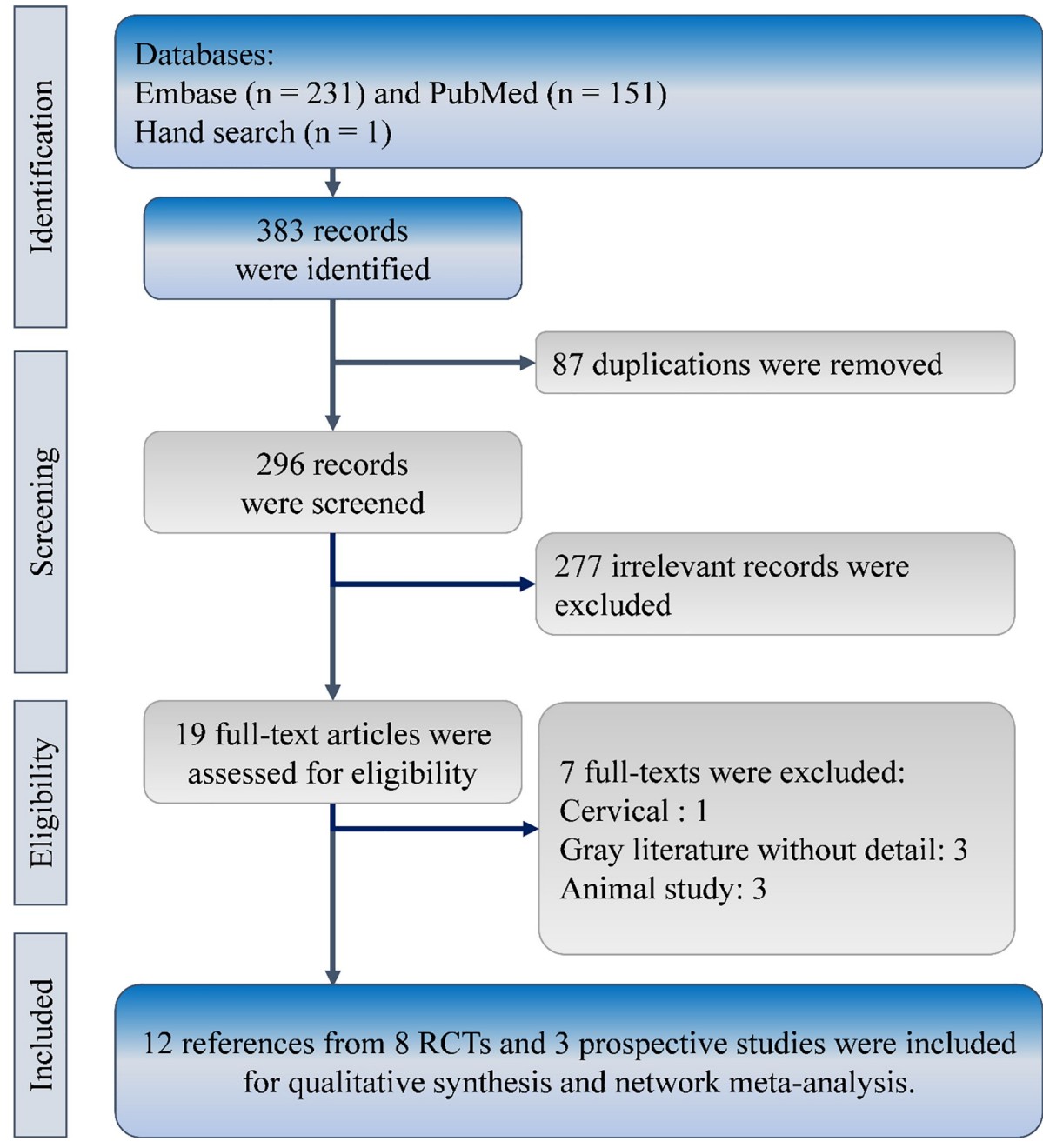

**Fig 1. Flow diagram of study selection.** RCT, randomized clinical trial.

### Oswestry disability index

Five studies reported data on ODI among teriparatide, bisphosphonate, and placebo (Fig 2B) [19, 22–25]. According to the available data, the network meta-analysis of ODI showed no significant differences among teriparatide, bisphosphonate, and placebo (S7 File in S1 Appendix). However, SUCRA indicated that teriparatide (Mean rank = 1.2; SUCRA = 88.2) may be a better treatment than bisphosphonate (Mean rank = 2.2; SUCRA = 39.8) and placebo (Mean rank = 2.6; SUCRA = 22.0; Fig 3B; S8 File in S1 Appendix). The loop inconsistency test for the

**Table 1. Characteristics of the included studies.**

| Author | Location | Study Design | Surgical indication | Group | Patient number | Age Mean (SD) | Sex (M/F) | Follow up duration |
|---|---|---|---|---|---|---|---|---|
| Jespersen 2019 | Denmark | RCT | Spondylolisthesis | Teriparatide | 41 | 71(1.01) | 11/30 | 12 months |
| | | | | Placebo | 46 | 70(0.88) | 7/39 | |
| Sheng 2018 | China | RCT | Spondylolisthesis HIVD, Spinal stenosis | Zoledronic acid | 28 | 60.7(6.2) | 7/21 | 12 months |
| | | | | null | 28 | 63.1(4.9) | 10/18 | |
| Ide 2018 | Japan | RCT | Spinal stenosis | Teriparatide + denosumab | 8 | 73.2(2.7) | 3/5 | 12 months |
| | | | | Teriparatide | 8 | 75.0(2.4) | 0/8 | |
| Seki 2017 | Japan | Prospective | Vertebral fracture | Teriparatide | 33 | 72.5(5) | 0/33 | 24 months |
| | | | | Alendronate/ risedronate | 25 | 71.5(2) | 0/25 | |
| Ebata 2017 | Japan | RCT | Lumbar degenerative disease | Teriparatide | 36 | 72.6(7) | 0/36 | 6 months |
| | | | | null | 38 | 70.4(8) | 0/38 | |
| Cho 2017 | Korea | Prospective | Spinal stenosis, spondylolisthesis | Teriparatide | 23 | 71.0(4.9) | 0/23 | 24 months |
| | | | | Alendronate | 24 | 68.2(8.4) | 0/24 | |
| Yagi 2016 | Japan | Prospective | Posterior long instrumented fusion | Teriparatide | 43 | 68.6(6.9) | 0/43 | 24 months |
| | | | | null | 33 | 66.7(6.9) | 0/33 | |
| Chen 2016 | China | RCT | Spondylolisthesis | zoledronic acid | 33 | 65(8) | 6/27 | 12 months |
| | | | | Saline | 36 | 63(7) | 7/29 | |
| Ohtori 2013 | Japan | RCT | Spondylolisthesis with spinal stenosis | Teriparatide | 20 | 78(6.0) | 0/20 | 12 months |
| | | | | Risedronate | 20 | 75(5.0) | 0/20 | |
| | | | | Control | 22 | 77(5.8) | 0/22 | |
| Li 2012 | China | RCT | Non-specific | Zoledronic acid | 28 | 63.63(6.36) | 13/28 | 12 months |
| | | | | Saline | 25 | 63.83(5.7) | 16/25 | |
| Nagahama 2011 | Japan | RCT | Spondylolisthesis and spinal stenosis | Alendronate | 19 | 70.3(8.6) | 1/18 | 12 months |
| | | | | null | 17 | 67.4(7.6) | 1/16 | |

HIVD, herniated intervertebral disc; RCT, randomized clinical trial.

network meta-analysis of ODI showed insignificance (chi-square = 2.22, *p* = 0.136; S9 File in S1 Appendix), and the Egger's test did not detect small study effects in this consistency model (*t* = 1.22, 95% CI: -2.36 to 6.65, *p* = .276; S10 File in S1 Appendix).

### Adverse event

Four of the eligible studies presented adverse event data on teriparatide, bisphosphonate, and placebo (Fig 2C) [17, 20, 23, 24]. The network meta-analysis of overall adverse event rate also showed insignificant differences among teriparatide, bisphosphonate, and placebo (S11 File in S1 Appendix). However, SUCRA still indicated that teriparatide (Mean rank = 1.1; SUCRA = 95.0) may be a better treatment than bisphosphonate (Mean rank = 2.4; SUCRA = 31.8) and placebo (Mean rank = 2.5; SUCRA = 23.2; Fig 3C; S12 File in S1 Appendix). The loop inconsistency test for the network meta-analysis of overall adverse event rate showed insignificance (chi-square = 0.06, *p* < .812; S13 File in S1 Appendix), and the Egger's test also detected no small study effects (t = -0.30, 95% CI: -2.85 to 2.29, *p* = .779; S14 File in S1 Appendix).

## Discussion

This study demonstrated higher fusion rate for the teriparatide group compared with the bisphosphonate and control groups. A trend of better clinical outcome and fewer adverse events

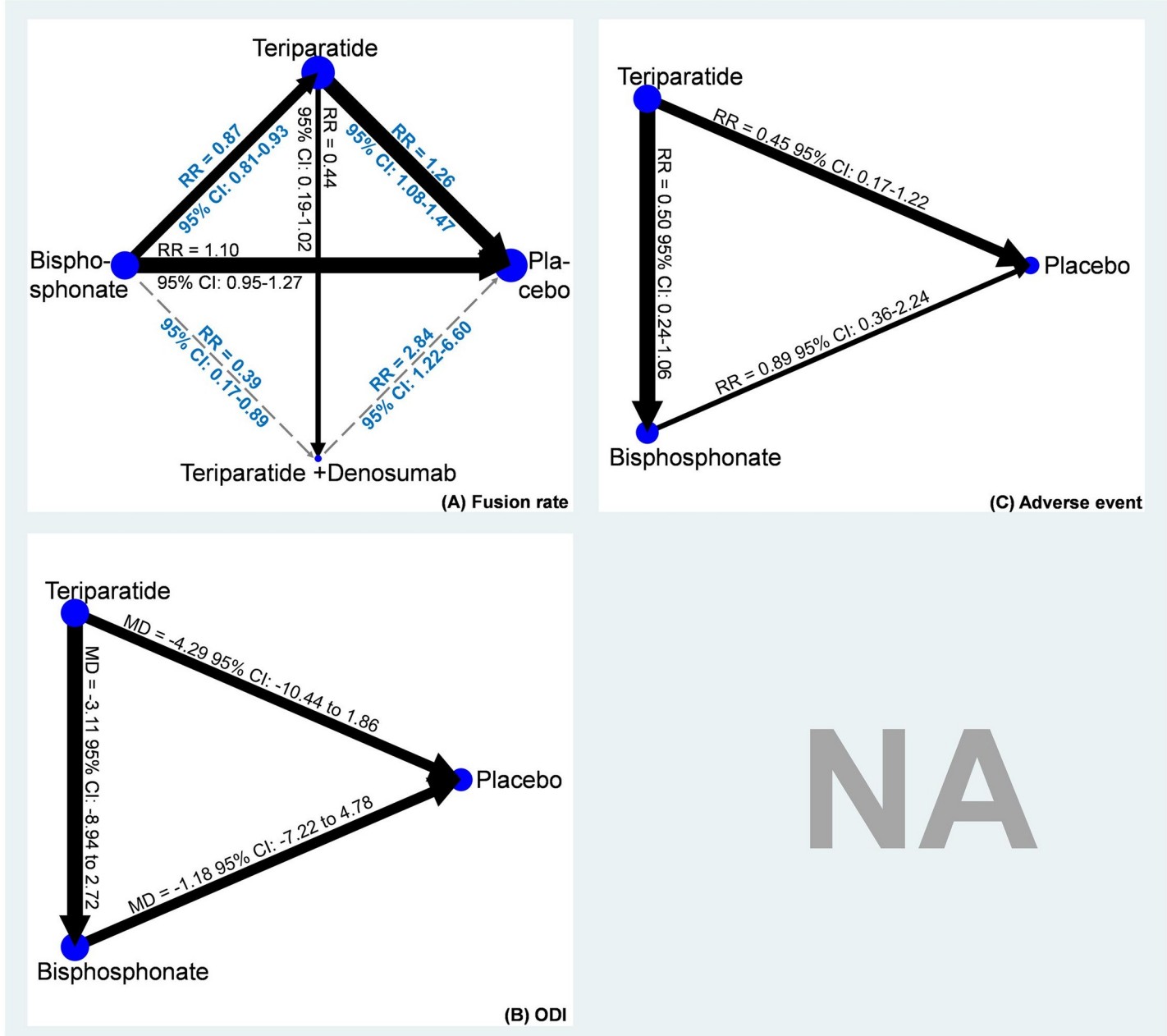

**Fig 2.** Network plots of (A) fusion rate, (B) Oswestry Disability Index (ODI), and (C) adverse event.

were also observed in the teriparatide group, though the difference did not reach statistical significance.

Teriparatide and bisphosphonate are commonly used anti-osteoporotic agents involving completely different mechanisms and having different impacts on bone metabolism. Teriparatide given intermittently increases bone anabolism and stimulates new bone formation [28, 29]. Bisphosphonate, on the contrary, is an anti-resorptive drug that suppresses the activity of osteoclast and increases bone mineral density [30, 31]. The present results of superior fusion rate achieved by teriparatide after spinal surgery compared with bisphosphonate can be

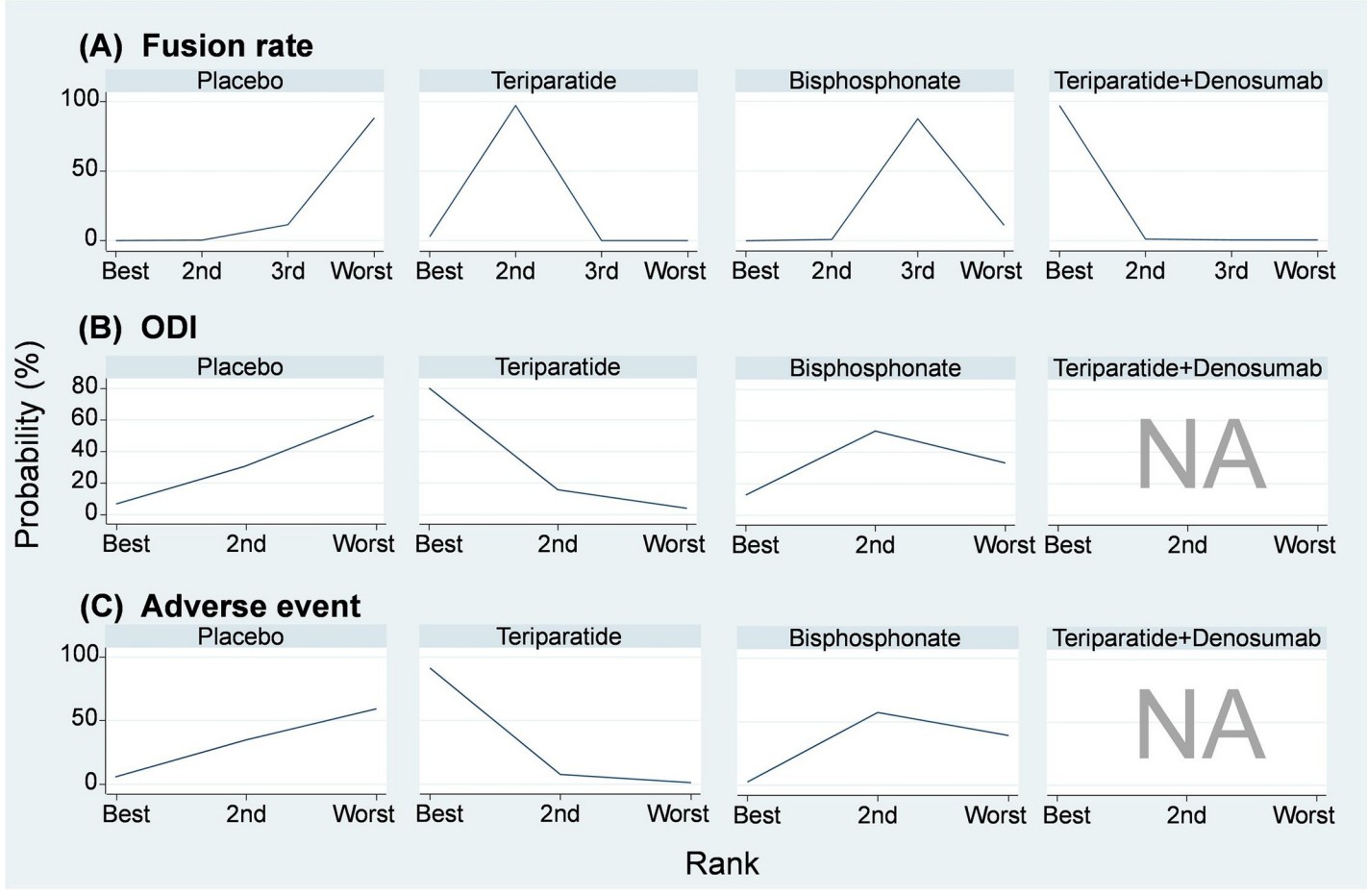

**Fig 3.** Probability rank of (A) fusion rate, (B) Oswestry Disability Index (ODI), and (C) adverse event.

explained by their different drug mechanisms. Although both drugs can increase the strength of vertebral body [32] and provide better support to the pedicle screws, only teriparatide can act on osteoblast and osteocyte to accelerate new bone formation. This anabolic effect may be the key that leads to unequal impact on intervertebral fusion mass formation, and can explain why the benefit of teriparatide was also observed among the non-osteoporotic population [20]. In previous studies, nonunion of fracture or arthrodesis has been treated using teriparatide with promising results [33–36]. Similar effect was observed in spinal fusion surgery.

Risk factor for pseudoarthrosis after spinal fusion included old age, large spondylolisthesis slip angle, infection, smoking and excessive motion at the fusion site [37–40]. In this study, fusion levels, fusion methods, fusion devices, graft selection, surgical indications and surgical methods all show heterogeneity which would influence fusion rate. Cervical spine surgery is not included in the analysis due to fundamental difference in anatomy, surgical approach, end-plate area and implant design in cervical spine, which should be discussed separately. Regarding diagnostic tools, there are different imaging modalities and grading systems for evaluating pseudoarthrosis [40, 41]. Currently, computed tomography and plain radiographies are most commonly used, but there are other tools including bone scan and positron emission tomography [40, 41]. The present review adopted stricter criteria for fusion; hence, "partial union" or "incomplete union" in the articles are not taken as solid fusion. Despite the abovementioned

heterogeneous factors, all results had I-square values less than 50%, representing acceptable heterogeneity.

There is still no consensus toward the dose of teriparatide and duration of use for stimulating spinal fusion. Most studies adhered to the dose of subcutaneous 20 µg daily for osteoporotic treatment [17, 19, 23, 24, 27]. However, the same dose given weekly in the study of Ebata et al. also achieved outcome superior to that of the control group [18]. A previous study combined denosumab, an antibody of receptor activator of nuclear factor kappa-B ligand (RANKL), with teriparatide and demonstrated better fusion rate. Although there are other works supporting the combined use of denosumab and teriparatide for the treatment of osteoporosis [42], clinical evidence is insufficient because only one single study used such regiment. There is insignificant difference in ODI among the three groups despite of variations in fusion rate. This is compatible with previous study that radiographic union is not associated with better short-term clinical result [43]. However, in the long run, solid union still guarantees better functional outcome [8].

This study is the first network meta-analysis that discussed the use of teriparatide and bisphosphonate after spinal fusion surgery. There are some other systemic reviews focusing on the same issue. Frestes et al. reported insignificant difference in fusion rate between bisphosphonate and control group but bisphosphonate could reduce cage subsidence and vertebral fracture [12]. On the contrary, meta-analysis of Liu et al. suggested that bisphosphonate may improve fusion rate [44], and the discrepancy may be due to inclusion of retrospective studies by Liu et al. A recent meta-analysis comparing teriparatide, bisphosphate and control simultaneously concluded that teriparatide achieved better fusion rate than bisphosphonate and control, while there is no difference between bisphosphonate and control [13]. However, retrospective studies were also included and network meta-analysis was not performed. Moreover, three new studies by Sheng, Ide et al. and Jespersen et al. were not included [19, 20, 25]. The present analysis demonstrated similar results that teriparatide was better than bisphosphonate and control on the basis of latest and higher quality evidences.

## Limitations

The study is limited by variations in follow-up duration and small number of included studies. Even without inconsistency and concerned heterogeneity detected, the evidence obtained may be not of high quality due to the mixed sample of randomized clinical trials and prospective comparative studies. Moreover, the consistency model of fusion rate used was an incomplete network meta-analysis. There was only one trial implementing combination of teriparatide and denosumab, and the estimates of the combination treatment relied only on this single trial. No randomized clinical trial or prospective comparative study comparing the combination therapy and bisphosphonate was found. To confirm the efficacy of teriparatide and bisphosphonate after spinal fusion, more well-designed randomized clinical trials on this topic with multiple arms should be examined. No recommendation on the dose of peri-operative teriparatide and duration of use can be given. Such would merit further exploration.

## Conclusion

This is the very first network meta-analysis provides an overview of the use of teriparatide and bisphosphonate for spinal fusion surgery. Teriparatide treatments for spinal fusion surgery can significantly improve fusion rate while bisphosphonate cannot. Teriparatide tends to improve clinical symptoms or decrease adverse events, but the differences do not reach statistical significance. Combined use of denosumab and teriparatide may result in better fusion rate

compared with using teriparatide alone, yet more evidence is necessary to support this combination therapy.

## Supporting information

**S1 Checklist.**
(DOC)

**S1 Appendix.**
(PDF)

## Author Contributions

**Conceptualization:** Yi-Jie Kuo.

**Data curation:** Shih-Hao Cheng.

**Formal analysis:** Yi-No Kang.

**Investigation:** Shih-Hao Cheng.

**Methodology:** Yi-No Kang.

**Supervision:** Yi-Jie Kuo, Chiehfeng Chen.

**Validation:** Yi-Jie Kuo, Chiehfeng Chen.

**Visualization:** Yi-No Kang.

**Writing – original draft:** Shih-Hao Cheng.

**Writing – review & editing:** Yi-Jie Kuo, Chiehfeng Chen, Yi-No Kang.

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
