## [Decision Letter · Decision Letter 0]

30 Jul 2020

Effects of teriparatide and bisphosphonate on spinal fusion procedure: a systematic review and network meta-analysis

PONE-D-20-09535

Dear Dr. Kang,

We’re pleased to inform you that your manuscript has been judged scientifically suitable for publication and will be formally accepted for publication once it meets all outstanding technical requirements.

Kind regards,

Svenja Illien-Jünger, Ph.D.

Academic Editor

PLOS ONE

Journal Requirements:

1. In the methods, please describe how risk of bias was assessed in individual studies (including specification of whether this was done at the study or outcome level, or both, and the specific test employed, such as the I^2 statistic), and how this information was used in any data synthesis.

Additional Editor Comments (optional):

Following the instructions of reviewer 2, please add osteopenia and osteoporosis as risk factors to the introduction.

Reviewers' comments:

Reviewer's Responses to Questions

**Comments to the Author**

1. Is the manuscript technically sound, and do the data support the conclusions?

Reviewer #1: Yes

Reviewer #2: Yes

2. Has the statistical analysis been performed appropriately and rigorously? 

Reviewer #1: Yes

Reviewer #2: Yes

3. Have the authors made all data underlying the findings in their manuscript fully available?

Reviewer #1: Yes

Reviewer #2: Yes

4. Is the manuscript presented in an intelligible fashion and written in standard English?

Reviewer #1: Yes

Reviewer #2: Yes

5. Review Comments to the Author

Reviewer #1: The authors present the first network meta-analysis (NWA) looking at the use of Teriparatide and bisphosphonates in the setting of spinal fusion surgery, which is an important topic with high real-world relevance. While other systematic reviews have been previously conducted in the literature, this is the first to use this newer NWA statistical method. Compared with traditional pairwise meta-analyses, NMAs allow the visualization of a larger amount of evidence, estimation of the relative effectiveness among all interventions, and rank ordering of the interventions. As such, they can be helpful in determining the best available intervention by facilitating indirect comparisons of multiple interventions that have not been robustly studied in head-to-head studies.

Based off their analyses, the authors conclude that Teriparatide treatment significantly improves spinal fusion rates while bisphosphonates do not. This conclusion is properly placed in the context of the previous literature and the author's provide a nice discussion comparing/contrasting their study to similar previously conducted systematic reviews on the subject matter. That being said, while their conclusion is supported by their data, there are many challenges with these types of analyses that can limit their real-world usefulness. For a NMA to produce valid results, it is important that the distribution of effect modifiers (i.e., average patient age, gender distribution) is similar, since this balance increases the plausibility of reliable findings from an indirect comparison. In my opinion, the authors did a nice job of providing this information systematically, which helps the reader empirically evaluate the validity of the assumption of transitivity by reviewing the distribution of potential effect modifiers across trials.

Overall, the manuscript is well organized and written clearly enough to be accessible to non-specialists and the details of their methodology are sufficient to allow the analyses to be reproduced.

Reviewer #2: Excellent paper on a very important topic. You list several risk factors for pseudoarthrosis, but do not list osteopenia and osteoporosis as risk factors, but several papers have shown that these are also independent risk factors as well. You may want to include that in the paper.

6. PLOS authors have the option to publish the peer review history of their article (what does this mean?). If published, this will include your full peer review and any attached files.

Reviewer #1: No

Reviewer #2: **Yes: **Ahmad Nassr

---

## [Editor Report · Acceptance letter]

5 Aug 2020

PONE-D-20-09535 

Effects of teriparatide and bisphosphonate on spinal fusion procedure: a systematic review and network meta-analysis 

Dear Dr. Kang:

I'm pleased to inform you that your manuscript has been deemed suitable for publication in PLOS ONE. Congratulations! Your manuscript is now with our production department. 

Kind regards, 

on behalf of

Dr. Svenja Illien-Jünger 

Academic Editor

PLOS ONE